# Simulation and Implementation of a Mobile Robot Trajectory Planning Solution by Using a Genetic Micro-Algorithm

**Jose Eduardo Cardoza Plata** [1] , **Mauricio Olguín Carbajal** [2], **Juan Carlos Herrera Lozada** [2],
**Jacobo Sandoval Gutierrez** [3,*] , **Israel Rivera Zarate** [2] **and Jose Felix Serrano Talamantes** [2]

[1]  Escuela Superior de Ingeniería Mecánica y Eléctrica (ESIME), Instituto Politécnico Nacional (IPN),
     Mexico City 02550, Mexico
[2]  Centro de Innovación y Desarrollo Tecnológico en Cómputo (CIDETEC), Instituto Politécnico Nacional (IPN),
     Mexico City 07700, Mexico
[3]  Departamento de Procesos Productivos, Universidad Autónoma Metropolitana (UAM),
     Lerma de Villada 52005, Mexico
*   Correspondence: j.sandoval@correo.ler.uam.mx

**Abstract:** Robots able to roll and jump are used to solve complex trajectories. These robots have a low level of autonomy, and currently, only teleoperation is available. When researching the literature about these robots, limitations were found, such as a high risk of damage by testing, lack of information, and nonexistent tools. Therefore, the present research is conducted to minimize the dangers in actual tests, increase the documentation through a platform repository, and solve the autonomous trajectory of a maze with obstacles. The methodology consisted of: replicating a scenario with the parrot robot in the gazebo simulator; then the computational resources, the mechanism, and the available commands of the robot were studied; subsequently, it was determined that the genetic micro-algorithm met the minimum requirements of the robot; in the last part, it was programmed in simulation and the solution was validated in the natural environment. The results were satisfactory and it was possible to create a parrot robot in a simulation environment analogous to the typical specifications. The genetic micro-algorithm required only 100 generations to converge; therefore, the demand for computational resources did not affect the execution of the essential tasks of the robot. Finally, the maze problem could be solved autonomously in a real environment from the simulations with an error of less than 10% and without damaging the robot.

**Keywords:** trajectory planning; mobile robot; micro-algorithm; genetic algorithm; simulation

## 1. Introduction

The technological variety of unmanned vehicles has grown with the development of ground and robotic aerial systems. On the one hand, robotic aerial systems tend to be faster, although they require more effort to control. On the other hand, terrestrial vehicles can move better in confined spaces, with the disadvantage of limited movements on elevated surfaces, as described by Tan et al. [1].

Studying this type of robot leads to the development of algorithms to make such movements autonomous by following an optimal trajectory. For example, in [1], Tan et al. use a fast exploration random tree method called RRT, a heuristic search algorithm based on sampling. Huizinga et al. [2] use a combinatorial multi-objective evolutionary algorithm, allowing all combinations of subtasks to be explored simultaneously. Such combinations are a series of steps to be followed for robot locomotion.

Using a three-layer neural network, Ramos et al. [3] have one of the three inputs connected to a sensor, an output connected to the actuators, and a hidden layer connecting the previous two layers. A simple evolutionary algorithm is also used to synthesize the control of the robot. Lan et al. [4] proposed a multi-objective trajectory planning method for collaborative robots; the higher-order-B-spline interpolation method is utilized to construct

a continuous trajectory. Then, a trajectory competitive multi-objective particle swarm optimization is used to optimize the joint trajectory.

The algorithms and others whose purpose is the autonomous motion of mobile robots generally explore a wide range of possible solutions simultaneously. Such exploration involves significant wear and tear on the physical robot, which is why a simulator is commonly used; with a simulator, an algorithm can be run and optimized as many times as necessary. This strategy was developed by Zhang et al. [5] for a wheeled robot employing an improved dolphin swarm algorithm on a real platform with the help of a simulation.

Another example is JBotEnvolver, a Java-based software with an open-source multi-robot simulation platform used by Ramos et al. [3]. Alternately, Huizinga et al. [2] and Dinev et al. [6] use PyBullet, a python module, to simulate robots utilizing a physics engine and machine learning.

Mulun Wu et al. in [7] used a simulation environment in a mixed reality implementing a modified vector field histogram (VFH) based routing algorithm as a path planning algorithm. The simulated environment and the modified VFH algorithm showed that this method proposed valid motion paths for the four-wheeled mobile robot concerning the operator's requirements for obtaining good obstacle avoidance performance.

Molina-Leal et al. [8] presented a long short-term memory (LSTM) neural network that allows a mobile robot to find the trajectory between two points and navigate while avoiding a dynamic obstacle. The authors used a LIDAR sensor to measure the distance between the robot and obstacles. The linear and angular velocity of the robot is obtained using a model to learn the mapping between input and output data. The mobile robot and its dynamic environment are simulated in Gazebo. The computer simulation showed that the network model could plan a safe navigation route in a dynamic environment.

In [1], Tan et al. use MATLAB to simulate the scenario and the proposed algorithm, solving the traveling salesman problem. Finally, in [9], Klemm et al. use Gazebo, an open-source 3D robotics simulator, to create a high-performance physics engine.

Other techniques are found in [10] and [11], where Hao et al. describe a path planning of mobile robots based on a multi-population migration in the genetic algorithm (see Figure 1).

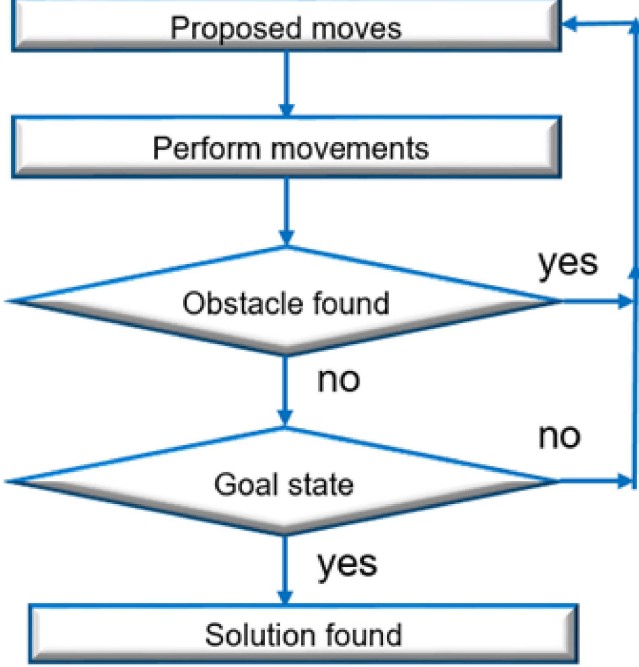

**Figure 1.** Flowchart for trajectory planning of mobile robots.

The heterogeneous contributions in the literature are classified into relevant metrics and compared, as shown in Table 1.

**Table 1.** Robot metrics comparison.

| Reference | Simulator Used/ Operating System | Embedded Algorithm | Implementation or Simulation | Algorithm Used | Time for Travel/Computing Time./Other |
|---|---|---|---|---|---|
| Tan et al. [1] | MATLAB/Windows | No, the solutions are computed on MATLAB | Both | Rapidly exploring random tree (RRT) | 36 s |
| Huizinga and Clune [2] | Bullet, Fastsim | No | Simulation | Multi-objective evolutionary algorithm (CMOEA) | It depends on the optimization result. |
| Ramos et al. [3] | JBotEvolver/Java-based open-source | No, the solutions are computed on software | Simulation | Continuous-time recurrent neural network | A simple obstacle (1.71 to 1.84 s) |
| Dinev et al. [6] | Simulator PyBullet | No, the solutions are computed using CasADI and KNITRO | Simulation | KNITRO and selected the Interior/Conjugate-Gradient algorithm suited for large-scale | 5 s |
| Klemm et al. [9] | Gazebo/ROS | Yes, the solution was implemented directly on the chip | Both | Optimal control strategy for regulating a linear system at minimal cost LQR | Time to jump 2 s |
| Chávez et al. [12] | MATLAB/Windows | Yes, in a P8X32A 8-core Architecture | Both | Genetic algorithm | Another type of robot |
| Lee et al. [13] | Java JDK1.8 / Windows 10 | No, the solutions are computed using CasADI and KNITRO | Simulation | Genetic algorithm (GA) and a direction factor toward a target point | Not presented. Only shows computing time 1.87 s |
| de Oliveira et al. [14] | MobileSim, Pioneer Software Development Kit to commercial Pioneer 3-DX robot./ROS. | No. Personal computer is responsible for acquiring the sensor data and controlling the robot. | Both | Hybrid path-planning strategy with A* algorithm | It depends on the optimization result |
| Hao et al. [11] | MATLAB r2018a / Windows 10(64-bit) | No, the solutions are computed in MATLAB | Simulation | Multi-population migration genetic algorithm | Not presented. Only shows computing time 120.59 s |
| Mengmei Liu [15] | MATLAB and SIMULINK | No, the solutions are computed on MATLAB. Do not use a mobile robot | Simulation | Genetic Algorithm | Time to do one simulation 2.5 s. total of simulation 1600 |
| Lan et al. [4] | MATLAB, ROS | No, the solutions are computed on MATLAB | Simulation | Multi-objective particle swarm optimization algorithm (TCMOPSO) | It depends on the optimization result |
| Zhang et al. [5] | none | Yes, the solution was implemented directly on the chip | Implementation | Dolphin swarm algorithm | 35 s time to trajectory |
| Mulun et al. [7] | ROS, Rviz/Windows | No, the solutions are computed on ROS | Both | Vector field histogram * (VFH *) | Not presented. Only shows computing time 62.2 s |
| Molina-Leal et al. [8] | GAZEBO/Windows | Both, TurtleBot 3 Waffle Pi | Both | Adam optimization algorithm | Another type of robot |
| Our proposed solution | ROS, GAZEBO, Linux. | Yes, the solution was implemented directly on the chip | Both | Micro-genetic algorithm | 33 s |

## 2. Materials and Methods

### 2.1. Robot

In this work, we use a two-wheeled mobile robot with jumping capability, a low-cost model called a Tuk-Tuk, and a multimodal locomotion robotic platform made by Parrot. This robot is only 15.5 cm wide and 14.3 cm long, with a height of 11.6 cm and a weight of 200 g (see Figure 2).

It can turn 360° in less than a second, reach a maximum speed of 13 km/h, and have a top jump of 80 cm.

The robot has a VGA camera with 640 x 480 pixels of resolution at 30 fps, an audio system for speaking and listening, 4 GB of internal memory, a 550 mAh average battery duration that allows us to use it for 20 min, and 2.5 GHz and 5 GHz Wi-Fi.

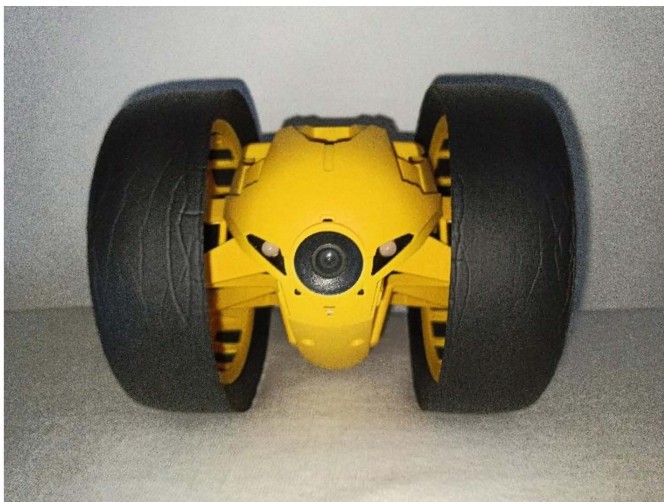

**Figure 2.** Tuk-tuk parrot robot.

*2.2. Algorithm*

For the robot's autonomous navigation, a bio-inspired algorithm was implemented, which is a heuristic technique whose behavior is inspired by the biological evolution of living beings and Neo-Darwinism. These algorithms are used in search, optimization, and learning procedures. Furthermore, these algorithms handle several potential solutions simultaneously as an alternative to commonly used techniques, which tend to search for solutions linearly.

Within these heuristic techniques, we find a set of algorithms whose model is obtained by applying the natural selection mechanism and the theories of evolution to population schemes; these techniques are known as evolutionary algorithms in [13,14,16].

These algorithms can be divided into three main stages:

- *Initialization:* an initial population of *n* individuals is randomly generated;
- *Generation:* in this stage, the corresponding genetic operators are applied to the initial population, depending on the paradigm being handled; they are also known as evolutionary adaptation procedures;
- *Cycle:* Finally, the second stage is repeated until the convergence criterion or stop condition is reached. Some algorithms cycle from the first stage, keeping the fittest individuals and replacing the rest of the population within a second convergence cycle.

Three basic genetic operators are used in the generation stage: selection, crossover (reproduction or recombination), and mutation;

- *Selection:* consists of a probabilistic or deterministic process that makes it possible to choose the parent individuals of the next generation;
- *Crossover:* refers to the exchange of information between two parents selected based on their fitness according to the objective function;
- *Mutation:* is responsible for making minimal changes to the newly created individuals in the new population to explore areas of the search space that the crossover could not reach, thus maintaining the diversity of the individuals.

Evolutionary algorithms use selection, mutation, and recombination to create diversity, and, according to Chávez [12] and Zhang et al. [17], are usually divided into three paradigms:

- Evolutionary programming;
- Evolutionary strategies;
- Genetic algorithms.

Currently, some trends combine characteristics of the three mechanisms and include techniques from other fields of study, such as search algorithms, machine learning, or data structures. This has given rise to paradigms such as the following:

- Differential evolution;
- Genetic programming;
- Memetic algorithm;
- Probabilistic models.

Genetic algorithms are used within the area of artificial intelligence in the field of optimization, manipulating simultaneously a set of potential solutions to a given problem that needs to be maximized or minimized.

In this work, we used a genetic algorithm with a reduced population, formally known as a genetic micro-algorithm. It obtains competitive results while requiring fewer resources than a standard algorithm. The genetic micro-algorithm is illustrated in Figure 3.

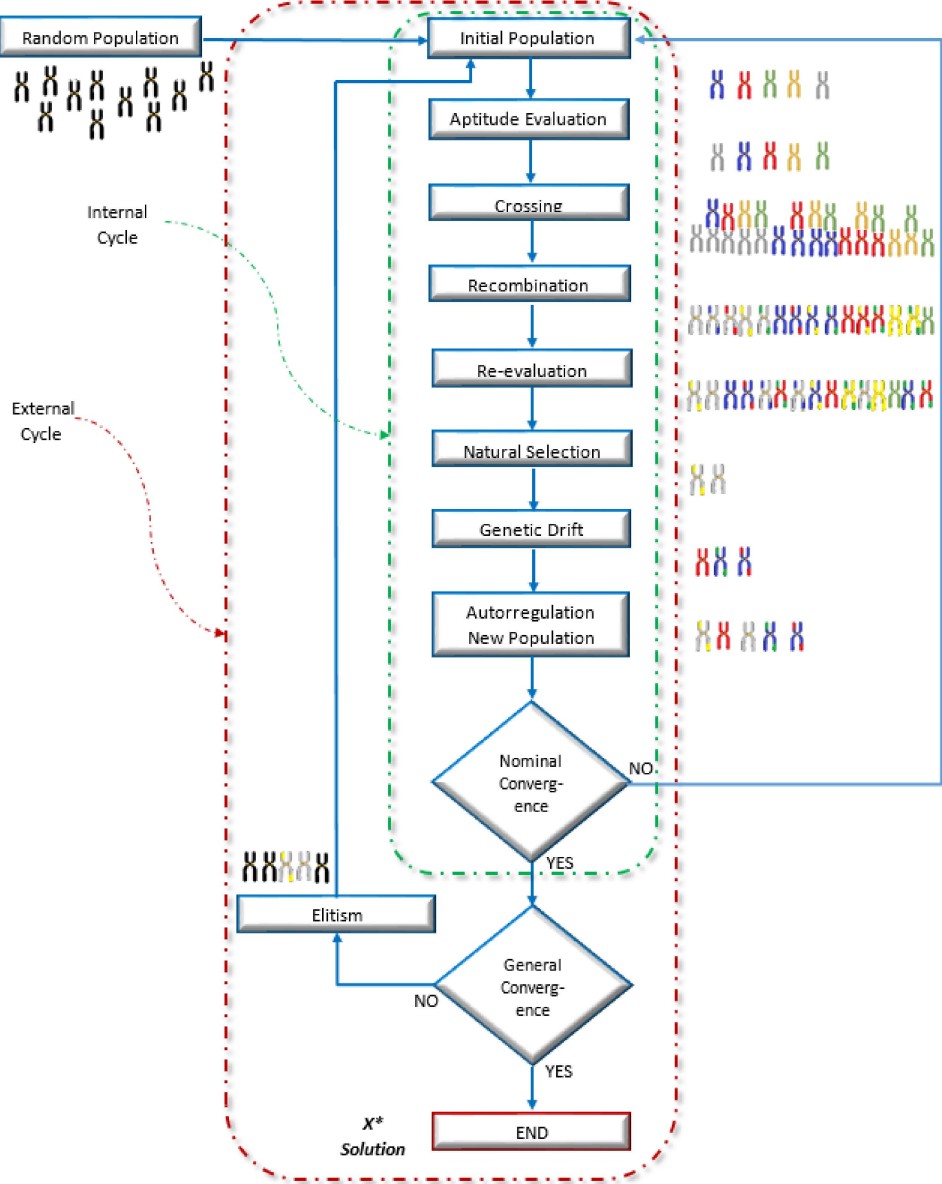

**Figure 3.** Genetic micro-algorithm fitness evaluation.

The algorithm begins generating a random population of five sets of movements, which are named individuals. Then, this population is evaluated based on its fitness.

The closer an individual is to the goal, the higher their fitness. Then, a crossing process is performed in which the fittest individuals have higher chances of being used for reproduction. They will also contribute the dominant alleles in the genetic recombination to their parents, forming both the parents and children of a new population, which is reevaluated by selecting the two best individuals (natural selection).

These individuals, together with three others selected randomly (genetic drift) will be the survivors of autoregulation, in which the population returns to its original size. The process is repeated cyclically until nominal convergence is reached. Then, through elitism, the two best individuals of the last population are selected.

These two individuals, together with three new randomly generated individuals are integrated into the new population that will appear at the beginning of the algorithm, thus forming a second loop with a stop condition established in the general convergence. At the end of this loop, we will have the best individual. This individual will result from genetic inheritance through several generations, and therefore, the best solution found.

The last stage, where new, totally random individuals are introduced, is known as stochastic noise. It helps to maintain diversity in the population, thus escaping premature convergence and avoiding being trapped in local optima. At the same time, elitism allows us to scale over the best solutions in search of a global optimum.

A balance must be maintained between the diversity provided, i.e., between the exploration and exploitation of the micro-algorithm. Chavez [12] mentions that exploitation is the process of using the information obtained from previously explored solutions to determine which solutions are more convenient to continue advancing toward, while exploration is the process of visiting new regions of the search space (new solutions) to see if something promising can be found.

*2.3. Simulator*

The open-source 3D robotics simulator Gazebo has a high-performance physics engine used to execute and evaluate each solution proposed by the genetic algorithm.

Gazebo uses a world description file in simulation description format (SDF). In this file, it is possible to specify all the physical characteristics of a robot and its movements, as well as the environment and lighting, among other objects.

To run the simulation, we use the Gazebo command followed by the name of the world file. This file is read by Gazebo, which starts a simulation based on what is described in the world file and ends the simulation when the user indicates that it should end.

The gzserver analyzes the world description file and performs the simulation without including a graphical interface. It can be executed from the command line and end when the user indicates it.

The gzclient takes care of the graphical interface by connecting to a running gzserver. It is worth mentioning that when using the Gazebo command, both gzserver and gzclient are executed together.

Gazebo allows us to save a log of the simulation in a file named state.log by adding the -r option on the command line.

The gzlog command allows us to filter a state.log file in such a way that we simplify the stored information according to the options specified by the user in the command line.

*2.4. Chromosome bit-to-motion conversion*

The Tuk-Tuk robot has two finite ranges of motion ($Motion = [Turn, Adv]$). The first motion is a turn in degrees, $Turn = [Turn_{min}, Turn_{max}]$. The second motion is a forward motion in cm, $Adv = [Adv_{min}, Adv_{max}]$. Each individual x has n subsets $S_i$ of $m_{odd}$ or $m_{even}$ bits in length ($x = S_{1,m} + S_{2,m} + \ldots + S_{n,m}$). Then it will represent the chromosome. For the movement, if the subset is odd, it corresponds Turn, and if even, it corresponds Adv. Then, we would have the execution of the robot's movement by $x_{motion} = Turn_1 + Adv_1 + \ldots + Turn_n + Adv_n$. Finally, to perform the conversion, we use the $Turn_i = \frac{Turn_{max} - Turn_{min}}{m_{odd}}$ or $Adv_i = \frac{Adv_{max} - Adv_{min}}{m_{even}}$.

For the case study, an individual was tested with a 144 bits chromosome with $n = 12$, $m_{odd} = 14bits$, $m_{even} = 10bits$, $Turn_{min} = -360°$, $Turn_{max} = 360°$, $Adv_{min} = 0$ cm, and $Adv_{max} = 50$ cm. The chromosome setup and an example are shown in Figure 4.

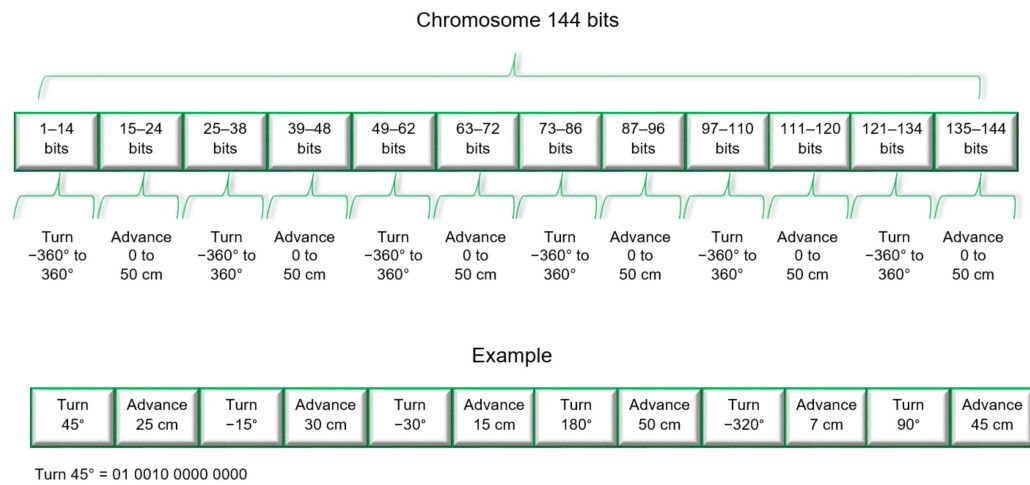

**Figure 4.** Chromosome-to-motion conversion.

### 2.5. Methods

The trajectories are generated offline and tested in two ways: The first method uses the Gazebo simulator, for which an extensive methodology is proposed that is like those used in other applications. The second method involves implementing the best trajectory in the robotic platform, which is intrinsically executed.

The integration of the genetic micro-algorithm with the simulator (see Figure 5) was done through a C language program executed in Ubuntu, through which the simulation configuration files for Gazebo (world) are read and written.

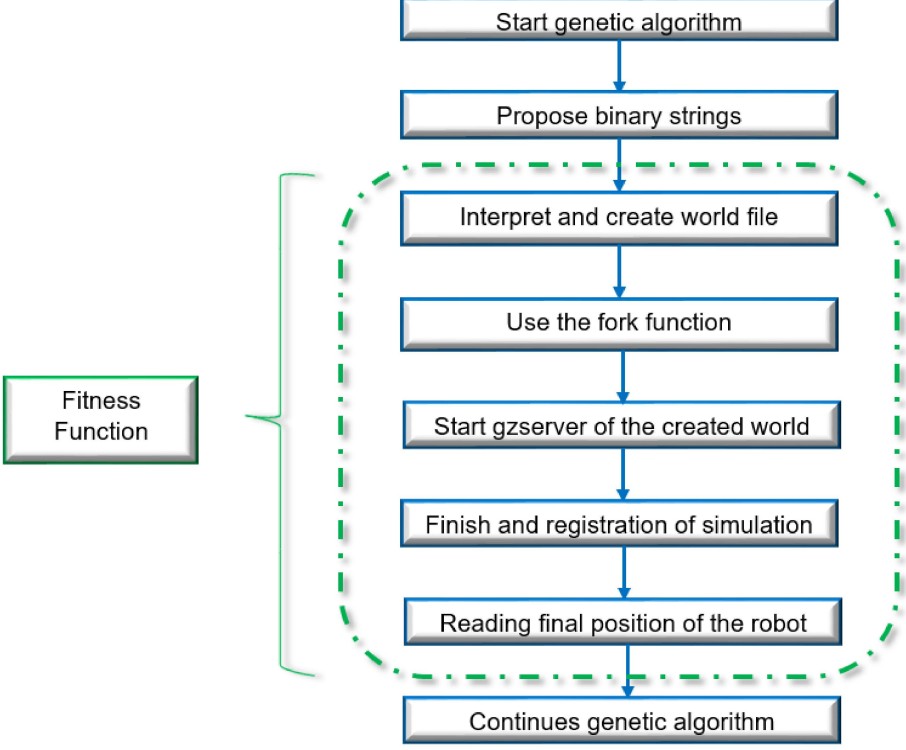

**Figure 5.** Integration of the genetic micro-algorithm with the simulator.

A world file contains all the necessary elements to simulate in Gazebo. For each set of movements, a world file is created. Then, it is simulated from the terminal with the gzserver command, which allows us to perform a simulation without using the graphical interface. This command is executed using the *fork*() function from the C program.

The *fork*() function creates a process, in this case, gzserver, and allows us to determine the duration of this process. Each simulation has a period depending on the number of movements of the robot.

Once the simulation process is finished, Gazebo will keep a record of it. Thus, from the C program, we execute the command *gzlog*, which allows us to interpret the description above and filter this record's information. Remember that we are interested in the final position of the robot concerning the established goal to evaluate the aptitude of the set of simulated movements.

The algorithm starts with the function *RandomPopulation*; Algorithm 1 is executed; it generates the initial population, a two-dimensional array whose dimensions are the population size times the size of the chromosomes. The population size is five, and each chromosome is a 144-bit binary string.

---

**Algorithm 1** RandomPopulation

---
       Input: populationSize, chromosome,
       sizeChormosome
       Output: Random Population
       1.   **function** RandomPopulation()
       2.   set RandomSeed()
       3.   **for** i ← 1 to *populationSize* **do**
       4.   RandomBinary(chromosome, sizeChromosome)
       5.   **end for**

---

Algorithm 2 shows the *RandomBinary* function used in line 4 of the *RandomPopulation* function, which pseudo-randomly generates a single 144-bit binary string.

---

**Algorithm 2** RandomBinary

---
       Input: sizeChromosome, chromosome
       Output: Random binary, chromosome
       1.   **function** RandomBinary()
       2.   **for** i ← 1 to *sizeChromosome* **do**
       3.      *RandomNum ← random* [1, 100]
       4.      **if** *RandomNum* ≤ 50 **do**
       5.         chromosome[ i ] ← 0
       6.      **else** chromosome[ i ] ← 1
       7.      **end if**
       8.   **end for**

---

In Algorithm 3, once the initial random population is created (line 1), the external cycle of the algorithm of thirty iterations starts (line 2), and the serial variable is set to 0 (line 3) in each iteration before starting the internal cycle (*nominalConv*), which consists of ten iterations (line 4).

At the beginning of the internal cycle (line 5), we generate a pseudo-random number from zero to five corresponding to the six movements of the robot. Then, this number is multiplied by 24, which is the number of bits occupied by each movement. Finally, the result of this multiplication is stored in the variable *fragment* (line 6).

In lines 7 to 10 of the main algorithm, a selection of the number of bits for the dynamic crossover is made. In the even iterations of the external cycle, this number will start at six and increase in each iteration of the internal cycle. For the odd iterations, it will start at 23 and will decrease in each iteration of the internal cycle. This number will be stored in the *serial* variable (lines 8 and 9).

---

**Algorithm 3** Main algorithm

---

Input: stopCondition, nominalConv, populationSize, chromosomeSize
Output: bestChromosome, resultTest1

1.  $P \leftarrow$ RandomPopulation()
2.  **for** $h \leftarrow 1$ to *stopCondition* **do**
3.      *serial* $\leftarrow 0$
4.      **for** i $\leftarrow 1$ to *nominalConv* **do**
5.          *rfrag* $\leftarrow$ random[0, 5]
6.          *fragment* $\leftarrow (24 \times rfrag)$
7.          **if** module(h+1) = 0 **do**
8.              *serial* $\leftarrow 6 + i$
9.          **else** *serial* $\leftarrow 23 - i$
10.         **end if**
11.         RandomSeed()
12.         FitnessEval($P$)
13.         *CrossPop* $\leftarrow$ Cross($P$)
14.         *SortedPop* $\leftarrow$ FitnessEval2(*CrossPop*)
15.         *NewPop* $\leftarrow$ SelfRegulation(*SortedPop*)
16.         **for** $j \leftarrow 1$ to *populationSize* **do**
17.             **for** $k \leftarrow 1$ to *chromosomeSize* **do**
18.                 $P[j][k] \leftarrow NewPop[j][k]$
19.             **end for**
20.         **end for**
21.     **end for**
22.     RandomBinary(sizeChromosome, $P[1]$)
23.     RandomBinary(sizeChromosome, $P[2]$)
24.     RandomBinary(sizeChromosome, $P[3]$)
25.     *inputFile* $\leftarrow$ openFile("resultsTest1.txt", "r")
26.     set $g \leftarrow 0$
27.     **while** *inputFile* $\neq$ EOF **do**
28.         *inputData* $\leftarrow$ readChar(*inputFile*)
29.         Aux[g] $\leftarrow$ *inputData*
30.         $g \leftarrow g + 1$
31.     **end while**
32.     closeFile(*inputFile*)
33.     *outputFile* $\leftarrow$ openFile("resultsTest1.txt", "w")
34.     set $e \leftarrow 0$
35.     **while** $e < g$ **do**
36.         *outputFile* $\leftarrow$ Aux[e]
37.         $e \leftarrow e + 1$
38.     **end while**
39.     *outputFile* $\leftarrow$ *bestFitness*
40.     closeFile(*outputFile*)
41.     *bestFile* $\leftarrow$ openFile("bestChromosome.txt","w")
42.     **for** i $\leftarrow 1$ to *chromosomeSize* **do**
43.         bestFile $\leftarrow P[populationSize\text{-}1][\,i\,]$
44.     **end for**
45.     closeFile(*bestFile*)
46. **end for**

---

We set the seed that generates the numbers pseudo-randomly as a function of time at the beginning of each inner cycle (line 11).

In line 12, *FitnessEval* function evaluates the initial population fitness and orders them from highest to lowest.

Subsequently, in line 13, we use the function *Cross*, which receives the initial population already evaluated and sorted and creates ten more chromosomes that, together with the initial population, will form the cross-population (*CrossPop*).

In line 14 of the main algorithm, we use *FitnessEval2*, which evaluates the cross-population and sorts it from highest to lowest fitness (*SortedPop*).

Then, in line 15, we use the *SelfRegulation* function, which selects the two best chromosomes from the sorted population (*SortedPop*) and randomly selects three different chromosomes, forming the new population (*NewPop*) of five chromosomes.

From lines 16 to 20, the new population (*NewPop*) is stored in the initial population (*P*), closing the internal cycle in line 21 of the main algorithm. Once this cycle is finished, the *RandomBinary* function is used to create three chromosomes in the last positions of the initial population (lines 22, 23, and 24). Note that the first two positions contain the two best chromosomes from the previous cycle.

In the following lines (25 to 32), we open the file named *resultsTest1*, save its content in an auxiliary array, add the best distance achieved in this iteration, and overwrite this file (lines 33 to 40). This action saves the progress of the algorithm over the generations. Finally, we overwrite the file *bestChromosome*, where we store the binary string of the best chromosome of this iteration (lines 41 to 45).

We close the outer loop at line 46, and the main algorithm concludes once the stop condition is reached.

In the function *FitnessEval*, in Algorithm 4, we go through the input population (line 2) and evaluate each of its chromosomes using the function *Fitness* (line 3), saving the result in another array (line 4), as well as saving the position of the corresponding chromosome (line 5). Subsequently, we use bubble sorting from lines 7–18 for the second array from highest to lowest fitness. Once sorted, we obtained the numbers of the corresponding chromosomes sorted and saved the chromosome in an auxiliary array (lines 19 to 24). Then, it is stored in the original array (lines 25 to 30).

---

**Algorithm 4.** FitnessEval

---

Input: populationSize, orderedPopulation
Output: orderedPopulation, bestFitness, P
1.　　**function** FitnessEval()
2.　　**for** $i \leftarrow 0$ to populationSize **do**
3.　　　　　　*fitness* $\leftarrow$ *FitnessFunction_call_to_gazebo_simulator*
4.　　　　　orderedPopulation[$i$][0] $\leftarrow$ *fitness*
5.　　　　　orderedPopulation[$i$][1] $\leftarrow$ *i*
6.　　**end for**
7.　　**for** $i \leftarrow 1$ to *populationSize*-1 **do**
8.　　　　　　**for** $j \leftarrow 0$ to *populationSize-i* **do**
9.　　　　　　　　**if** *orderedPopulation[j][0] > orderedPopulation[j+1][0]* **do**
10.　　　　　　　　　　*aux* $\leftarrow$ *orderedPopulation[j][0]*
11.　　　　　　　　　　*orderedPopulation[j][0]* $\leftarrow$ *orderedPopulation[j+1][0]*
12.　　　　　　　　　　*orderedPopulation[j+1][0]* $\leftarrow$ *aux*
13.　　　　　　　　　　*aux* $\leftarrow$ *orderedPopulation[j][1]*
14.　　　　　　　　　　*orderedPopulation[j][1]* $\leftarrow$ *orderedPopulation[j+1][1]*
15.　　　　　　　　　　*orderedPopulation[j+1][1]* $\leftarrow$ *aux*
16.　　　　　　　　**end if**
17.　　　　　　**end for**
18.　　**end for**
19.　　**for** $i \leftarrow 0$ to *populationSize* **do**
20.　　　　　*k* $\leftarrow$ *orderedPopulation[i][1]*
21.　　　　　**for** $j \leftarrow 0$ to *chromosomeSize* **doend for**
22.　　　　　　　*aux*[i][j] $\leftarrow$ *population*[k][j]
23.　　　　　**end for**
24.　　**end for**
25.　　**for** $i \leftarrow 0$ to *populationSize* **do**
26.　　　　　**for** $j \leftarrow 0$ to *chromosomeSize* **do**
27.　　　　　　　*P[populationSize-i-1]* $\leftarrow$ *aux[i][j]*
28.　　　　　**end for**
29.　　　　　*bestFitness* $\leftarrow$ *orderedPopulation[i][0]*
30.　　**end for**

The fitness function is described in Equation (1).

$$F(x_i) = G - x_i \tag{1}$$

where G is the goal and $x_i$ is the final position obtained for each individual simulation where i = 1, 2, 3, ..., n.

## 3. Results

For the experimental stage, the genetic micro-algorithm was used with binary representation, with populations of five individuals. Each individual is a binary string of 144 bits.

The robot can perform 360° turns to the right or left, using 14 bits for each turn. It can also move forward 50 cm, which requires 10 bits.

It follows that each individual will be able to perform six turns and six advances interspersed with each other, i.e., after a turn it performs an advance and after an advance it performs a turn. A mutation was implemented for each bit of (1/chain length) × 10.

A dynamic crossover process was implemented, varying between 6 and 23 bits per string fragment throughout the generations. Each fragment is made up of 24 bits, which represent a turn and an advance. Two of them are crossed randomly in each process.

The robot starts at point (0, 0) and must move over a stage with obstacles until it reaches the goal, which is 2 m ahead of the starting point, has its center at (2, 0), and an area of 20 cm. Table 2 shows the parameters for matching virtual and real orientations and Table 3 shows the parameters for matching virtual distances with real distances.

**Table 2.** Turning parameters.

| Measured | Simulation | Robot |
|:---:|:---:|:---:|
| 360° | 358°–361° | 354°–366° |
| 180° | 179°–181° | 173°–184° |
| 90° | 89°–91° | 86°–94° |
| 45° | 43°–46° | 43°–48° |
| 10° | 9°–11° | 8°–12° |
| 5° | 4°–7° | 3°–8° |

**Table 3.** Advancing parameters.

| Measured | Simulation | Robot |
|:---:|:---:|:---:|
| 1 m | 98–102 cm | 94–107 cm |
| 50 cm | 48–52 cm | 45–55 cm |
| 25 cm | 24–27 cm | 21–28 cm |
| 15 cm | 14–16 cm | 12–18 cm |
| 10 cm | 9–11 cm | 8–12 cm |
| 5 cm | 3–7 cm | 2–7 cm |

The first experiment consists of a scenario with a single obstacle between the goal and the starting point; this obstacle is 50 cm from the starting point (see Figure 6).

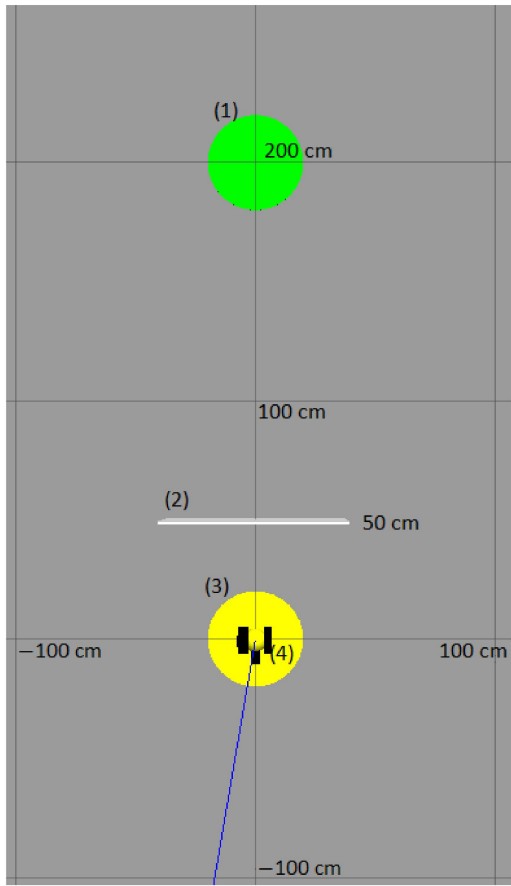

**Figure 6.** Simulated scenario with one obstacle: (1) goal, (2) obstacle, (3) home, and (4) robot.

The real implementation of this scenario is shown in Figure 7.

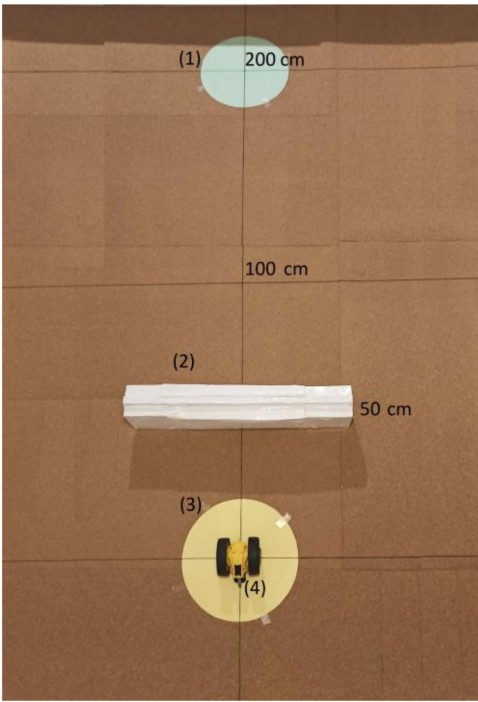

**Figure 7.** Scenario with one obstacle: (1) goal, (2) obstacle, (3) home, and (4) robot.

After three hundred generations, the best movement combination was obtained. These movements make up the trajectory shown in Figure 8.

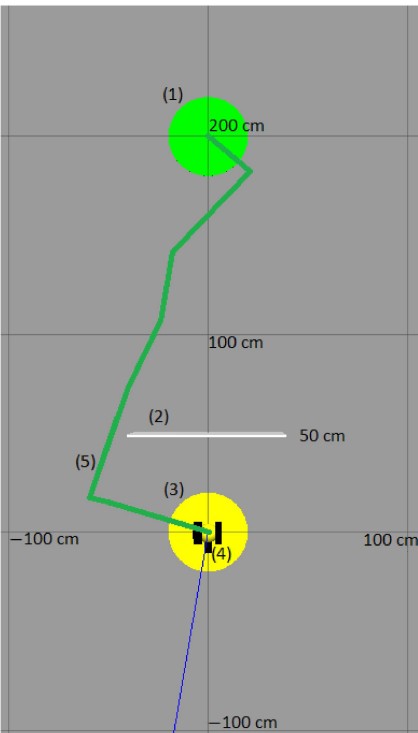

**Figure 8.** Trajectory in simulated scenario with one obstacle: (1) goal, (2) obstacle, (3) home, (4) robot, and (5) trajectory.

The robot executed these movements using a similar trajectory to reach the goal (see Figure 9).

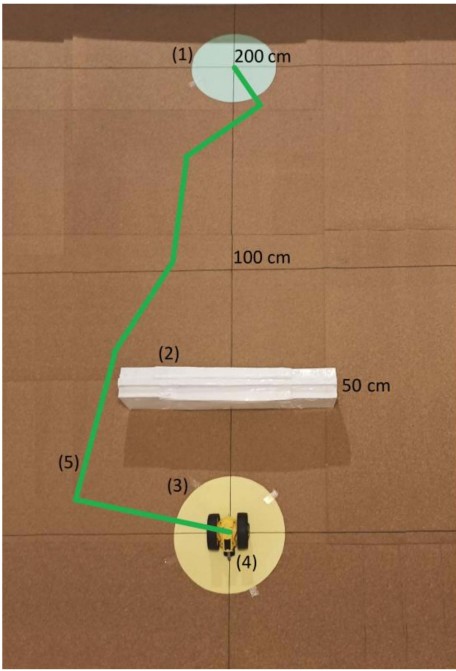

**Figure 9.** Trajectory in scenario with one obstacle: (1) goal, (2) obstacle, (3) home, (4) robot, and (5) trajectory.

The graph in Figure 10 allows us to appreciate the algorithm's evolution through the generations, which gives better results as time passes. The goal, or optimal result, is reached in the 120th generation.

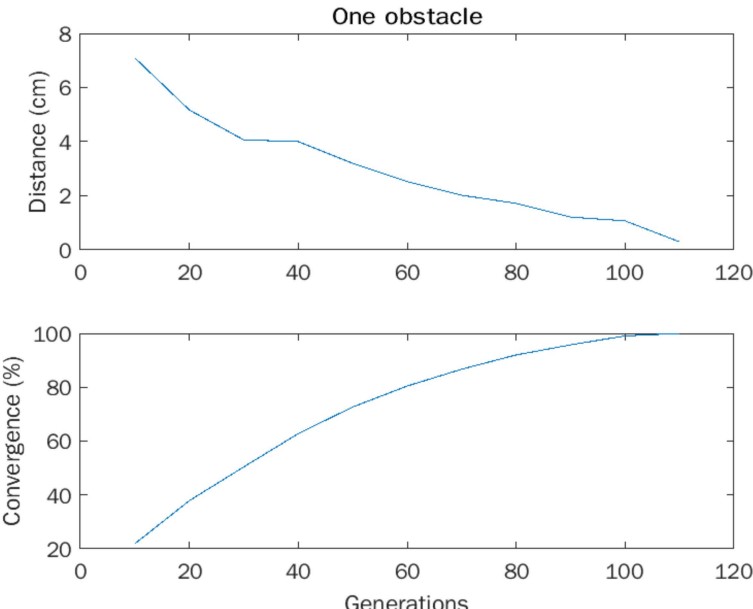

**Figure 10.** Results of experiment 1.

The second obstacle is added to the stage 100 cm from the goal in the second experiment. Each obstacle is 75 cm long, starting from the center-right and left, respectively (see Figure 11), so the robot can avoid them both in the middle and on the sides.

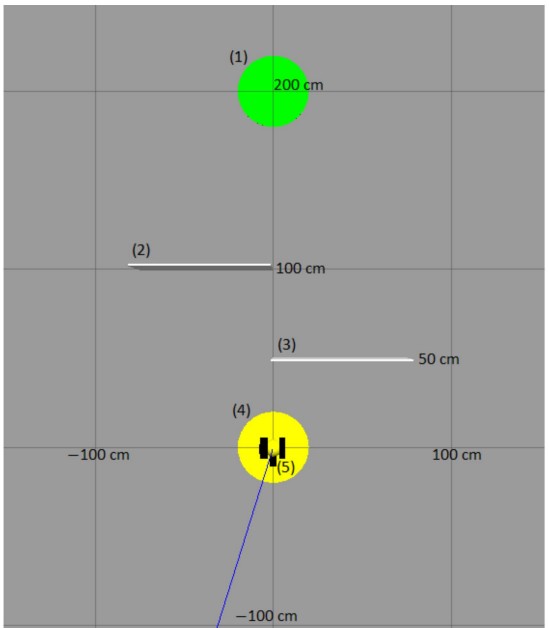

**Figure 11.** Simulated scenario with two obstacles: (1) goal, (2, 3) obstacles, (4) home, and (5) robot.

The physical implementation of this scenario is shown in Figure 12.

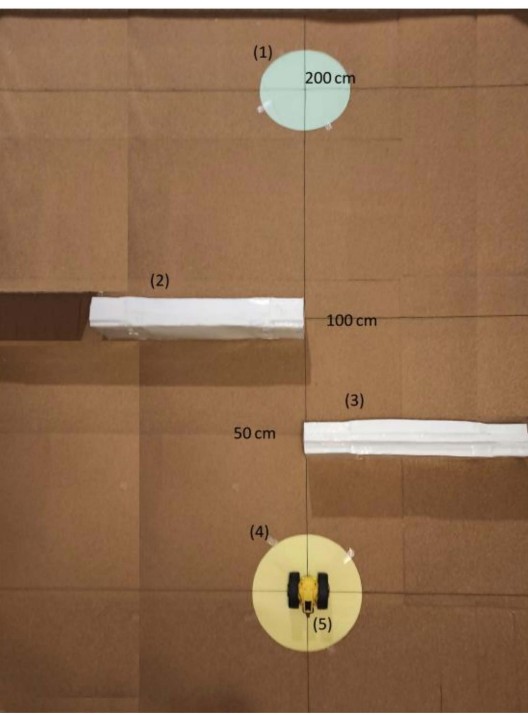

**Figure 12.** Scenario with two obstacles: (1) goal, (2,3) obstacles, (4) home, and (5) robot.

The best motion proposal obtained at the end of 300 generations is simulated. It describes the trajectory shown in Figure 13.

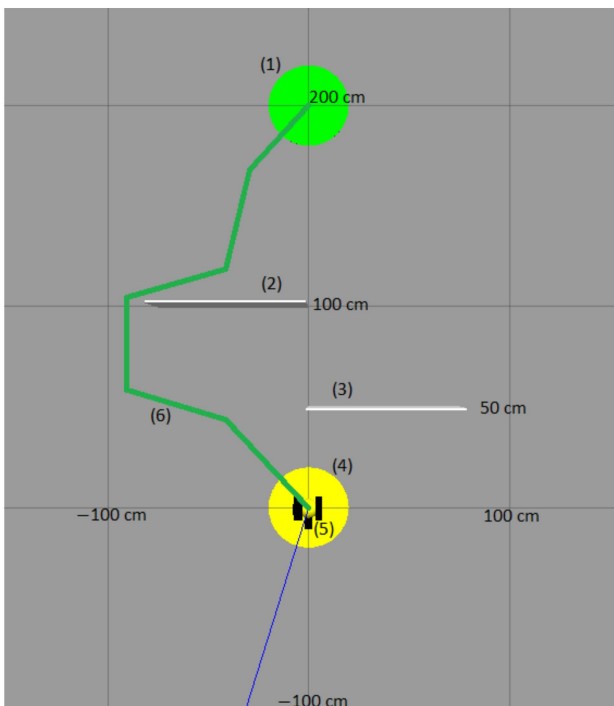

**Figure 13.** Trajectory in simulated scenario with two obstacles: (1) goal, (2,3) obstacles, (4) home, (5) robot, and (6) trajectory.

The robot executed the proposed movements, which used a similar trajectory to reach the intended goal (see Figure 14).

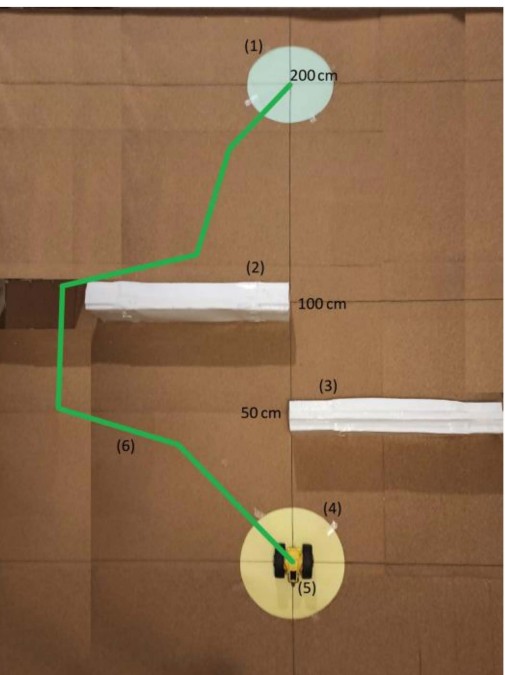

**Figure 14.** Trajectory in a scenario with two obstacles: (1) goal, (2, 3) obstacles, (4) home, (5) robot, and (6) trajectory.

The behavior of the algorithm in a scenario with two obstacles can be seen in Figure 15. Compared to the first experiment, in which close approximations to the goal are achieved in a few generations, more generations are required to achieve better results when a second obstacle is added.

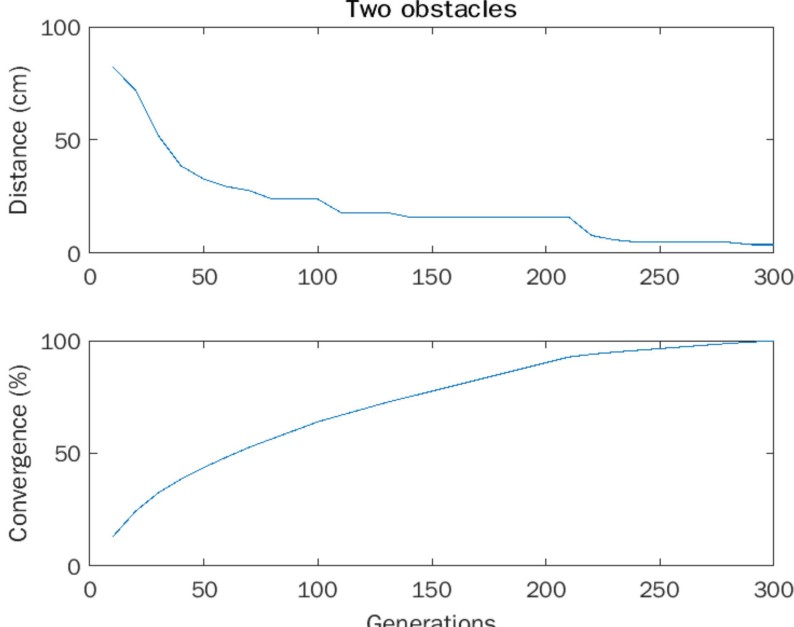

**Figure 15.** Results of experiment 2.

The third experiment consists of a scenario composed of a closed maze with two obstacles. The starting point and the goal are enclosed in the maze, forcing the robot to negotiate the obstacles to create a successful trajectory (see Figure 16).

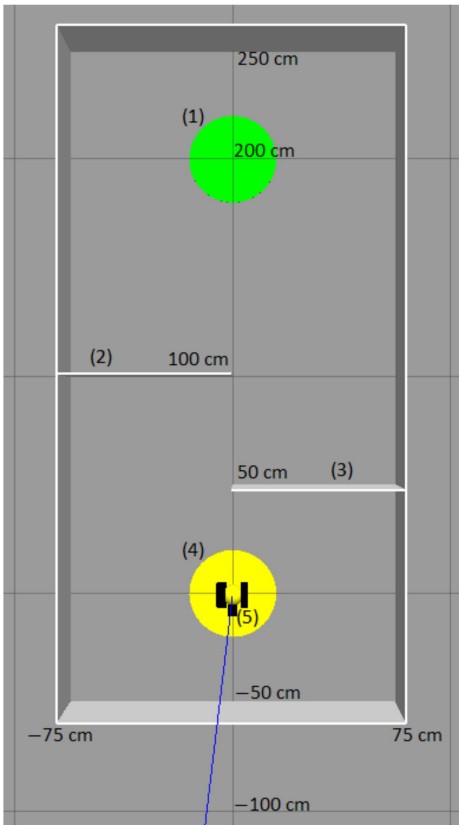

**Figure 16.** Simulation of labyrinth with two obstacles: (1) goal, (2,3) obstacles, (4) home, and (5) robot.

We can observe the physical representation of the maze with two obstacles in Figure 17.

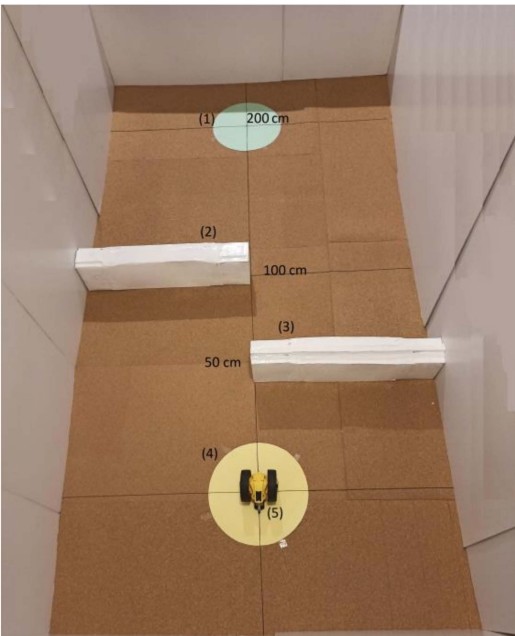

**Figure 17.** Maze with two obstacles: (1) goal, (2,3) obstacles, (4) home, and (5) robot.

After 300 generations, the best motion proposal obtained is simulated; it describes the trajectory shown in Figure 18.

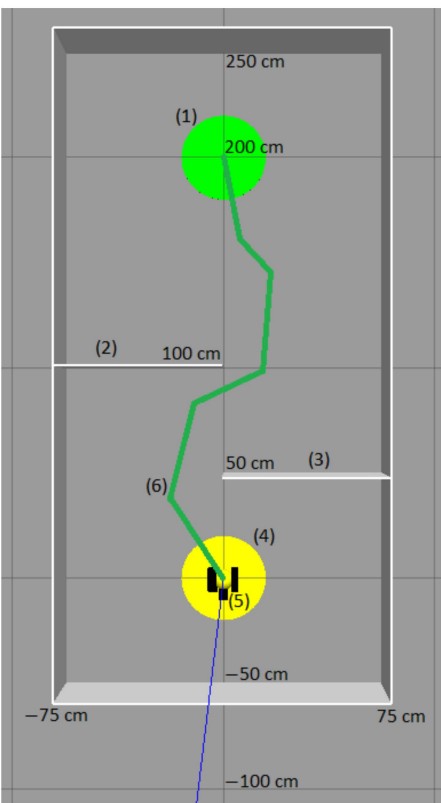

**Figure 18.** Trajectory of the maze with two simulated obstacles: (1) goal, (2,3) obstacles, (4) home, (5) robot, and (6) trajectory.

We can observe the results after 300 generations in Figures 19 and 20, where the best-fitting solution results in the robot being 7 cm from the center of the goal, so it locates the robot within the desired area.

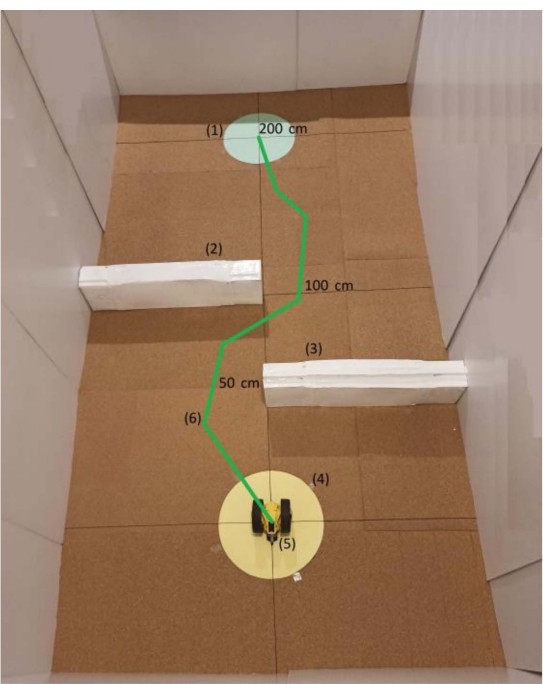

**Figure 19.** Trajectory in a labyrinth with two obstacles: (1) goal, (2,3) obstacles, (4) home, (5) robot, and (6) trajectory.

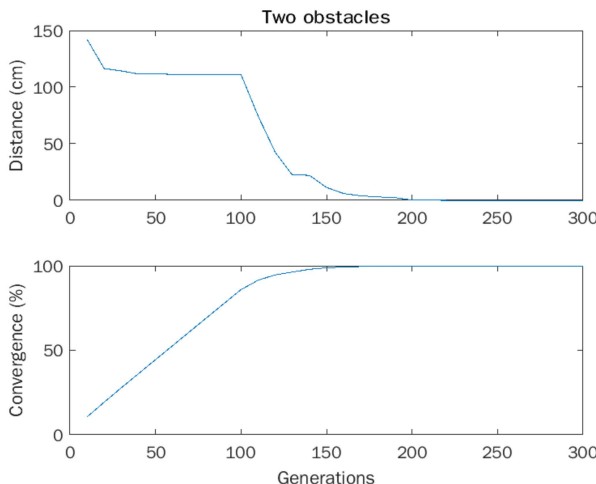

**Figure 20.** Results of the maze with two obstacles.

We can observe the error graph of the algorithm: zero, the ideal distance, was reached in generation 200, so 300 generations were not required. We can also observe a reduction of the error in the first 150 generations; the distance to the goal decreased from 200 cm to 11 cm, within the tolerance margin of 20 cm. Then, the algorithm reduces the margin until reaching the minimum error of 0 in generation 220.

Table 4 shows the results of ten experiments performed for 300 generations, considering the execution time and the target achieved.

**Table 4.** Time and goal results.

| Experiment | Generations | Time (min) | Goal (µm) |
|:---:|:---:|:---:|:---:|
| 1 | 300 | 1920.0 | 040 |
| 2 | 300 | 1920.0 | 100 |
| 3 | 300 | 1920.6 | 000 |
| 4 | 300 | 1920.0 | 200 |
| 5 | 300 | 1928.4 | 100 |
| 6 | 300 | 1926.0 | 100 |
| 7 | 300 | 1923.0 | 020 |
| 8 | 300 | 1920.0 | 010 |
| 9 | 300 | 1920.0 | 000 |
| 10 | 300 | 1924.2 | 000 |

Table 5 shows the descriptive statistics, and we can observe that both the compute time and goal error follow a normal distribution. This stochastic problem was estimated by a limited number of random samples [15].

**Table 5.** Time and goal results.

| Date | Time (min) | Goal (µm) |
|:---:|:---:|:---:|
| Maximum | 1928.40 | 200 |
| Minimum | 1920.00 | 000 |
| Mode | 1920.00 | 000 |
| Median | 1922.22 | 057 |
| Standard deviation | 0002.88 | 062 |

The behaviors for time and target are shown in Figure 21.

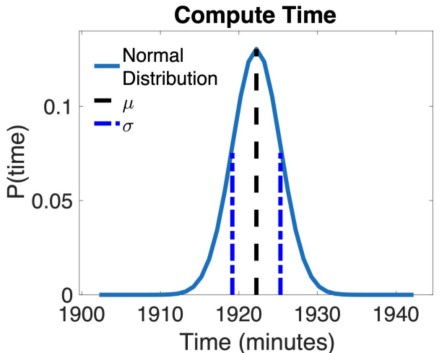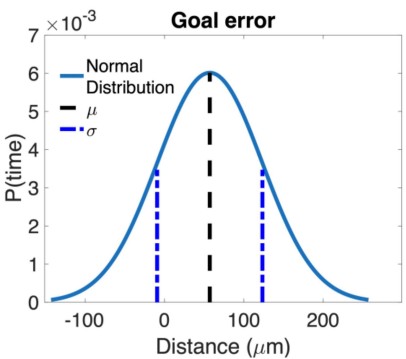

**Figure 21.** Time and goal results.

A recording of the algorithm performance, the simulation, and the real robot movement can be found at https://youtu.be/i2rkelWQpQ4 (accessed on 6 November 2022).

## 4. Discussion

It is possible to generate a simulation configuration file (world) for each proposed solution from a binary chain conceived by the genetic micro-algorithm. Likewise, the simulation process is managed by employing the *fork*() function, which determines the simulation's start, end, and duration, allowing us to run simulations automatically.

Some indirect advantages of using simulations are the access to multiple complementary tools that the robots would not have physically. For example, saving the simulation step by step in memory would allow analyzing parameter performances.

The robot crosses the trajectory in 33 s, successfully solving the maze; this result is acceptable if we compare it with other algorithms used in similar studies, as is the case of [1], where the robot crosses the trajectory on a stage with obstacles in a time of 36 s. The documentation and the repository code are available at the following link https://github.com/EduardoCardoza/Micro-AG-for-mobile-robot (accessed on 6 November 2022).

## 5. Conclusions

The objective of this work was achieved. In general terms, this development uses the Gazebo robotic simulator, a suitable genetic algorithm, and implements the best trajectory in a physical autonomous mobile robot. The main conclusions obtained are the following:

In the Gazebo simulator environment, a virtual model of a two-wheeled robot is possible. A set of scenarios were modeled for the robot to travel from a starting point to a predetermined destination without anomaly behavior. The simulator discovers failed tests in the virtual world to focus only on the best trajectory planning for the real world.

The parrot robot represents a challenge because it has a closed-embedded system; this limitation of resources and commands was solved with a heuristic technique. Implementing algorithms that are intrinsically executed in embedded systems is a current trend that refers to using architectures with limited resources, so the genetic micro-algorithm presented in this work was implemented.

Searching for the best solution from a wide range of potential solutions is difficult for the algorithm. The key to obtaining good performance is to have a functional chromosome bit-to-motion conversion relationship, a correct fitness function, the size of individuals, and the number of operations. Specifically, in the first experiments performed, an improvement in the solutions is observed across generations, reducing the distance between the mobile robot and the goal. The convergence graphs show good results after 100 generations and successful results after 200 generations. In the final experiments, there were good results at 150 generations and successful results at 300 generations. Other scenarios with more obstacles or distance toward the goal require more generations and execution time to solve the path in the algorithm.

The wear and tear also were a critical criterion to validate the proposed work. As a result of the parrot-robot physical implementation, an average of 10% relative error was obtained. This percentage is enough to avoid partial or total damage to our robot to achieve the goal of distances up to 2 m and few obstacles. A public test bed virtual repository is available for interested readers who wish to prove different scenarios or make adaptations.

At this stage, we could move the parrot robot wheels to avoid obstacles in a simulated and natural environment maze. For future work, we propose to improve the micro-genetic algorithm. First, we are modifying the proposed micro-algorithm to generate the trajectory of a jumping parrot robot considering an environment with dynamic obstacles, and second, several virtual robots together in a collaborative scheme.

**Author Contributions:** Conceptualization, M.O.C.; Formal analysis, J.C.H.L. and J.S.G.; Investigation, J.E.C.P. and M.O.C.; Methodology, J.C.H.L.; Project administration, J.C.H.L.; Resources, I.R.Z. and J.F.S.T.; Software, J.E.C.P.; Validation, J.E.C.P. and J.F.S.T.; Visualization, I.R.Z.; Writing—original draft, M.O.C. and J.S.G.; Writing—review & editing, J.S.G. All authors have read and agreed to the published version of the manuscript.

**Funding:** This research received no external funding, and the APC was funded by all authors.

**Conflicts of Interest:** The authors declare no conflict of interest.

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
