# Peer review of "Simulation and Implementation of a Mobile Robot Trajectory Planning Solution by Using a Genetic Micro-Algorithm"

_applsci, doi:10.3390/app122111284_

Round 1
Reviewer 1 Report
- The abstract is difficult to read and understand, it should be rewritten.
- Table-1 and table-2 are not referenced in the text. It is not explained how the values in the table should be read.
- The results of the proposed algorithm in the discussion section should be compared with the algorithms used in similar studies in the literature. This section is not enough and should be rewritten.
- Conclusion section also is difficult to read and understand, it should be rewritten.
Author Response
Response to Reviewer 1 Comments
Point 1: The abstract is difficult to read and understand, it should be rewritten.
Response 1: We rewrote the abstract.
Robots able to roll and jump are used to solve complex trajectories. These robots have a low level of autonomy, and currently, only teleoperation is available. When researching the literature about these robots, limitations were found, such as a high risk of damage by testing, lack of information, and nonexistent tools. Therefore, the present research was conducted to minimize the dangers in actual tests, increase the documentation through a platform repository, and solve the autonomous trajectory of a maze with obstacles. The methodology consisted of replicating a scenario with the parrot robot in the gazebo simulator; the computational resources, the mechanism, and the available commands of the robot were studied; subsequently, it was determined that the genetic micro-algorithm met the minimum requirements of the robot; and in the last part, it was programmed in simulation, and the solution was validated in the natural environment. The results were satisfactory; It was possible to create a parrot robot in a simulation environment analogous to the specifications. The genetic micro-algorithm required only 100 generations to converge; therefore, the demand for computational resources did not affect the execution of the essential tasks of the robot. Finally, the maze problem could be solved autonomously in a natural environment from the simulations with an error of less than 10% and without damaging the robot..
Point 2: Table-1 and table-2 are not referenced in the text. It is not explained how the values in the table should be read. (317 and 318 lines)
Response 2: We referenced the tables in the text and changed the captions (330 to 336 lines)
Table 1 shows the parameters for matching virtual and real orientations and Table 2 shows the parameters for matching virtual distances with real distances. (Table 1. Turning parameters. and Table 2. Advancing parameters.)
Point 3: The results of the proposed algorithm in the discussion section should be compared with the algorithms used in similar studies in the literature. This section is not enough and should be rewritten.
Response 3: We compared the solution with the reference [1] and rewrote part of this section
Some indirect advantages of using simulations are the access to multiple complementary tools that the robots would not have physically. For example, saving the simulation step by step in memory would allow analyzing parameter performances.
The robot crosses the trajectory in a time of 33 s successfully solving the maze; this result is acceptable if we compare it with other algorithms used in similar studies, as is the case of [1], where the robot crosses the trajectory on a stage with obstacles in a time of 36 s. The documentation and the repository are available at the following link https://github.com/EduardoCardoza/Micro-AG-for-mobile-robot.
Point 4: Conclusion section also is difficult to read and understand, it should be rewritten.
Response 4: We rewrote the conclusion section
The objective of this work was achieved. In general terms, this development uses the Gazebo robotic simulator, a suitable genetic algorithm, and implements the best trajectory in a physical autonomous mobile robot. The main conclusions obtained are the following:
In the Gazebo simulator environment, a virtual model of a two-wheeled robot is possible. A set of scenarios were modeled for the robot to travel from a starting point to a predetermined destination without anomaly behavior. The simulator discovers failed tests in the virtual world to focus only on the best trajectory planning for the real world.
The parrot robot represents a challenge because it has a closed-embedded system; this limitation of resources and commands was solved with a heuristic technique. Implementing algorithms that are intrinsically executed in embedded systems is a current trend that refers to using architectures with limited resources, so the genetic micro-algorithm presented in this work was implemented.
Searching for the best solution from a wide range of potential solutions is difficult for the algorithm. The key to obtaining a good performance is to have a functional chromosome bit-to-motion conversion relationship, a correct fitness function, the size of individuals, and the number of operations. Specifically, In the first experiments performed, an improvement in the solutions is observed across generations, reducing the distance between the mobile robot and the goal. The convergence graphs show good results after 100 generations and successful results after 200 generations. In the final experiments, there were good results at 150 generations and successful results at 300 generations. Other scenarios with more obstacles or distance toward the goal require more generations and execution time to solve the path in the algorithm.
The wear and tear also were a critical criterion to validate the proposed work. As a result of the parrot-robot physical implementation, an average of 10% relative error was obtained. This percentage is enough to avoid partial or total damage to our robot to achieve the goal of distances up to 2 meters and few obstacles. A public test bed virtual repository is available for interested readers who wish to prove different scenarios or make adaptations.
In this project stage, we could move the parrot robot wheels to avoid obstacles in a simulated and natural environment maze. For future work, two works are open. First, we are modifying the proposed micro algorithm to generate the trajectory of a jumping parrot robot considering an environment with dynamic obstacles, and second, several virtual robots together in a collaborative scheme.

Reviewer 2 Report
The paper tries to solve the optimal trajectory planning problem in robotics by using the family Genetic algorithm techniques with reduced population. The paper is well written in terms of describing how to use micro-genetic problem to achieve the task with well defined results and supporting discussions. Here are some of the improvements that can be made to the paper:
- The related work is not sufficiently enough to cover various approaches presented in the literature.
- The comparison among the related work is not provided that will make the efforts more valuable. Genetic algorithms are usually slow algorithms, another approach might be using reinforcement learning algorithm to achieve the same and do some comparison. (The use of deep learning to make robot learn about the path in an unknown environment.)
- The other factor that should be included is the scale of the algorithm in terms of time complexity in case number of obstacles increases and distance to the goal increases as well. How does the algorithm scale and whether it is useful to use it? Or this might be an overkill for complex paths.
- The future work is not well defined. adding to the repo is not a future work that will improve the state of the art. may be how to scale it, or what if we want to extend this technique to four wheel system, what will be the changes and improvements required to the system, can become future work.
Author Response
Response to Reviewer 2 Comments
Point 1: The related work is not sufficiently enough to cover various approaches presented in the literature..
Response 1: We rewrote the section and added references.
. Lan et al. [14] proposed a multi-objective trajectory planning method for collaborative robots; the higher-order-B-spline interpolation method is utilized to construct a continuous trajectory. Then a trajectory competitive multi-objective particle swarm optimization is used to optimize the joint trajectory.
The algorithms and others whose purpose is the autonomous motion of mobile robots generally explore a wide range of possible solutions simultaneously. Such exploration involves significant wear and tear on the physical robot, which is why a simulator is commonly used; with a simulator, an algorithm can be run and optimized as many times as necessary. This strategy was developed by Zhang et al. [15] for a wheeled robot employing an improved dolphin swarm algorithm on a real platform with the help of a simulation.
Another example is JBotEnvolver, Java-based software with an open-source multi-robot simulation platform used by Ramos et al. [3]. Huizinga et al. [2] and Dinev et al. [4] use PyBullet, a python module, to simulate robots utilizing a physics engine and machine learning.
Mulun Wu et al. in [16] used a simulation environment in a mixed reality implementing a modified vector field histogram (VFH) based routing algorithm as a path planning algorithm. The simulated environment and the modified VFH algorithm showed that this method proposed valid motion paths for the four-wheeled mobile robot concerning the operator's requirements for obtaining good obstacle avoidance performance.
Molina-Leal et al. [17] presented a Long Short-Term Memory (LSTM) neural network that allows a mobile robot to find the trajectory between two points and navigate while avoiding a dynamic obstacle. The authors used a LIDAR sensor to measure the distance between the robot and obstacles. The linear and angular velocity of the robot is obtained using a model to learn the mapping between input and output data. The mobile robot and its dynamic environment are simulated in Gazebo. The computer simulation showed that the network model could plan a safe navigation route in a dynamic environment.
- Lan, J.; Xie, Y.; Liu, G.; Cao, M. A Multi-Objective Trajectory Planning Method for Collaborative Robot. Electronics 2020, 9, 859. https://doi.org/10.3390/electronics9050859
- Zhang, X.; Huang, Y.; Rong, Y.; Li, G.; Wang, H.; Liu, C. Optimal Trajectory Planning for Wheeled Mobile Robots under Localization Uncertainty and Energy Efficiency Constraints. Sensors 2021, 21, 335. https://doi.org/10.3390/s21020335
- Mulun Wu,Shi-Lu Dai and Chenguang Yang“Mixed Reality Enhanced User Interactive Path Planning for Omnidirectional Mobile Robot”. 2020. Applied Sciences Journal, Ed. MDPI, Appl. Sci. 2020, 10, 1135; https://doi:10.3390/app10031135
Point 2: The comparison among the related work is not provided that will make the efforts more valuable. Genetic algorithms are usually slow algorithms, another approach might be using reinforcement learning algorithm to achieve the same and do some comparison. (The use of deep learning to make robot learn about the path in an unknown environment.)
Response 2: These robots have a low level of autonomy, and currently, only teleoperation is available. When researching the literature about these robots, limitations were found, such as a high risk of damage by testing, lack of information, and nonexistent tools. The parrot robot represents a challenge because it has a closed-embedded system; this limitation of resources and commands was solved with a heuristic technique. Implementing algorithms that are intrinsically executed in embedded systems is a current trend that refers to using architectures with limited resources, so the genetic micro-algorithm presented in this work was implemented. We compared the performance with a similar work.
The robot crosses the trajectory in a time of 33 s successfully solving the maze; this result is acceptable if we compare it with other algorithms used in similar studies, as is the case of [1], where the robot crosses the trajectory on a stage with obstacles in a time of 36 s. The documentation and the repository are available at the following link https://github.com/EduardoCardoza/Micro-AG-for-mobile-robot.
Point 3: The other factor that should be included is the scale of the algorithm in terms of time complexity in case number of obstacles increases and distance to the goal increases as well. How does the algorithm scale and whether it is useful to use it? Or this might be an overkill for complex paths.
Response 3: Searching for the best solution from a wide range of potential solutions is difficult for the algorithm. The key to obtaining a good performance is to have a functional chromosome bit-to-motion conversion relationship, a correct fitness function, the size of individuals, and the number of operations. Specifically, In the first experiments performed, an improvement in the solutions is observed across generations, reducing the distance between the mobile robot and the goal. The convergence graphs show good results after 100 generations and successful results after 200 generations. In the final experiments, there were good results at 150 generations and successful results at 300 generations. Other scenarios with more obstacles or distance toward the goal require more generations and execution time to solve the path in the algorithm. The documentation and the repository are available at the following link https://github.com/EduardoCardoza/Micro-AG-for-mobile-robot. The complex paths is a future work.
Point 4: The future work is not well defined. adding to the repo is not a future work that will improve the state of the art. may be how to scale it, or what if we want to extend this technique to four-wheel system, what will be the changes and improvements required to the system, can become future work.
Response 4: In this project stage, we could move the parrot robot wheels to avoid obstacles in a simulated and natural environment maze. For future work, two works are open. First, we are modifying the proposed micro algorithm to generate the trajectory of a jumping parrot robot considering an environment with dynamic obstacles, and second, several virtual robots together in a collaborative scheme.

Round 2
Reviewer 2 Report
After the extension of conclusion and related work including fine improvements throughout the paper, paper covers the relevant details that are required to demonstrate the effectiveness of the approach proposed.
It would be better if authors could have created a comparison table to list all important metrics based on which they have compared their approach with the related work. this would have given better picture about the literature work. this would tell readers clearly what are the tradeoffs and performance of different approaches.
Author Response
Response to Reviewer 2 Comments
Point 1: It would be better if authors could have created a comparison table to list all important metrics based on which they have compared their approach with the related work. this would have given better picture about the literature work. this would tell readers clearly what are the tradeoffs and performance of different approaches.
Response 1: We add a comparison table
|
Reference |
Simulator used / Operating System |
Embedded algorithm |
Implementation or Simulation |
Algorithm used |
Time for Travel / Computing time. / Other |
|
Tan et al. [1] |
MATLAB / Windows |
No, the solutions are computed on Matlab |
Both
|
Rapidly-exploring random tree (RRT*) |
36 s
|
|
Huizinga and Clune [2] |
Bullet, Fastsim |
No |
Simulation |
Multi-Objective Evolutionary Algorithm (CMOEA |
It depends on the optimization result.
|
|
Ramos et al. [3] |
JBotEvolver / a Java-based open-source |
No, the solutions are computed on software |
Simulation |
Continuous-time recurrent neural network |
A simple obstacle (1.71 to 1.84 s) |
|
Dinev et al. [4] |
Simulator PyBullet |
No, the solutions are computed using CasADI and KNITRO |
Simulation |
KNITRO and selected the Interior/Conjugate-Gradient algorithm suited for large-scale |
5s |
|
Klemm et al. [5] |
Gazebo/ROS |
Yes, the solution was implemented directly on the chip |
Both |
optimal control strategy for regulating a linear system at minimal cost LQR |
Time to jump 2s |
|
Chávez et al. [7] |
MATLAB / Windows |
Yes, in a P8X32A 8-core Architecture |
Both. |
Genetic algorithm |
Other type of robot |
|
Lee et al. [8] |
Java JDK1.8 / Windows 10 |
No, the solutions are computed using CasADI and KNITRO |
Simulation only |
Genetic algorithm (GA) and a direction factor toward a target point |
Not presented. Only Shows computing time 1.87 s |
|
de Oliveira et al. [10] |
MobileSim, Pioneer Software Development Kit to commercial Pioneer 3-DX robot. / ROS. |
No. Personal computer is responsible for acquiring the sensor data and controlling the robot. |
Both |
Hybrid Path-Planning Strategy with A* algorithm |
It depends on the optimization result.
|
|
Hao et al.[12] |
Matlab r2018a / Windows 10(64-bit) |
No, the solutions are computed on Matlab |
Simulation only |
Multi-population migration genetic algorithm |
Not presented. Only Shows computing time 120.59 s |
|
Mengmei Liu [13] |
MATLAB and SIMULINK |
No, the solutions are computed on Matlab. Don’t use a mobile robot |
simulation |
Genetic Algorithm |
Time to do one simulation 2.5s. total of simulation 1600. |
|
Lan et al. [14] |
MATLAB, ROS |
No, the solutions are computed on Matlab |
Simulation |
Multi-objective particle swarm optimization algorithm (TCMOPSO) |
It depends on the optimization result.
|
|
Zhang et al. [15] |
none |
Yes, the solution was implemented directly on the chip |
implementation |
dolphin swarm algorithm |
35s time to trajectory |
|
Mulun et al. [16] |
ROS, Rviz / Windows |
No, the solutions are computed on ROS. |
Both |
Vector field histogram* (VFH*) |
Not presented. Only Shows computing time 62.2 s |
|
Molina-Leal et. al, [17] |
GAZEBO / WINDOWS |
Both, TurtleBot 3 Waffle Pi |
Both |
Adam optimization algorithm |
Other type of robot |
|
Our proposed solution |
ROS, GAZEBO, UNIX |
Yes, the solution was implemented directly on the chip. |
Both |
Micro-genetic Algorithm |
33 s |
